# Treatment of Classic Mid-Trimester Preterm Premature Rupture of Membranes (PPROM) with Oligo/Anhydramnion between 22 and 26 Weeks of Gestation by Means of Continuous Amnioinfusion: Protocol of a Randomized Multicentric Prospective Controlled TRIAL and Review of the Literature

**DOI:** 10.3390/life12091351

**Published:** 2022-08-30

**Authors:** Michael Tchirikov, Christian Haiduk, Miriam Tchirikov, Marcus Riemer, Michael Bergner, Weijing Li, Stephan Henschen, Michael Entezami, Andreas Wienke, Gregor Seliger

**Affiliations:** 1Clinic of Obstetrics and Prenatal Medicine, Center of Fetal Surgery, University Hospital Halle (Saale), Martin-Luther-University Halle-Wittenberg, 06120 Halle (Saale), Germany; 2Center of Clinical Studies, Martin Luther University Halle-Wittenberg, 06108 Halle (Saale), Germany; 3Clinic of Obstetrics and Gynecology, St. Joseph Krankenhaus Berlin Tempelhof, 12101 Berlin, Germany; 4Clinic of Obstetrics and Gynecology, Hamburg Medical School, Helios Clinics GmbH, 19049 Schwerin, Germany; 5Center of Prenatal Diagnostic and Human Genetic, 10719 Berlin, Germany; 6Institute of Medical Epidemiology, Biostatistics and Informatics, Martin-Luther-University Halle-Wittenberg, 06120 Halle (Saale), Germany

**Keywords:** amnioinfusion, anhydramnion, preterm premature rupture of membranes, PPROM, neonatal survival, Amnion Flush Method

## Abstract

Background: The classic mid-trimester preterm premature rupture of membranes (PPROM) is defined as a rupture of the fetal membranes prior to 28 weeks of gestation (WG) with oligo/anhydramnion; it complicates approximately 0.4–0.7% of all pregnancies and is associated with very high neonatal mortality and morbidity. Antibiotics have limited success to prevent bacterial growth, chorioamnionitis and fetal inflammation. The repetitive amnioinfusion does not work because fluid is lost immediately after the intervention. The continuous amnioinfusion through the transabdominal port system or catheter in patients with classic PPROM shows promise by flushing out the bacteria and inflammatory components from the amniotic cavity, replacing amniotic fluid and thus prolonging the PPROM-to-delivery interval. Objective: This multicenter trial aims to test the effect of continuous amnioinfusion on the neonatal survival without the typical major morbidities, such as severe bronchopulmonary dysplasia, intraventricular hemorrhage, cystic periventricular leukomalacia and necrotizing enterocolitis one year after the delivery. Study Design: We plan to conduct a randomized multicenter trial with a two-arm parallel design. Randomization will be between 22/0 and 26/0 SSW. The control group: PPROM patients between 20/0 and 26/0 WG who will be treated with antibiotics and corticosteroids (from 22/0 SSW) in accordance with the guidelines of German Society of Obstetrics and Gynecology (standard PPROM therapy). In the interventional group, the standard PPROM therapy will be complemented with the Amnion Flush Method, with the amnioinfusion of Amnion Flush Solution through the intra-amnial catheter (up to 100 mL/h, 2400 mL/day). Subjects: The study will include 68 patients with classic PPROM between 20/0 and 26/0 WG. TRIAL-registration: ClinicalTrials.gov ID: NCT04696003. German Clinical Trials Register: DRKS00024503, January 2021.

## 1. Introduction

Mid-trimester PPROM complicates approximately 0.4–0.7% of all pregnancies and is associated with very high neonatal mortality and severe morbidity [1,2,3,4]. The causes of the mid-trimester PPROM are multifactorial [4]. The mechanism of PPROM is discussed as altered membrane morphology, including marked swelling and disruption of the collagen network, which can be triggered by bacterial products and pro-inflammatory cytokines and involves the activation of matrix metalloproteinases [4,5,6,7].

The propagation of bacteria is an important contributing factor in adverse neonatal and maternal outcomes after PPROM [1,4,8]. Gomez et al. reported that antibiotics failed to eliminate the amniotic infection in 83% of PPROM cases [9]. The “classic PPROM” with oligo/anhydramnion is associated with a short latency period and worse neonatal outcome compared to similar gestational aged neonates [4].

In Germany, the number of pregnant women with mid-trimester classic PPROM has been calculated to be about 5 thousand–7 thousand per year. Generally, about 40% of very preterm infants who survive the initial NICU die during next 5 years of life, and the long-term morbidity of the survivors remain high. Furthermore, more than 40% of surviving neonates following PPROM prior to 25 weeks of gestation develop bronchopulmonary dysplasia (BPD) later on [4,10].

The **fetal inflammatory response syndrome (FIRS)**, a systemic inflammation determined by elevated fetal plasma IL-6 concentrations and histological signs of funisitis in the umbilical cord, is an additional independent risk factor for the occurrence of severe neonatal morbidity [11].

The lavage effect of continuous amnioinfusion could protect the fetus and amniotic cavity from bacterial colonization; reduce the inflammatory response; and thus protect the neonate from major complications, such as pulmonary hypoplasia, sepsis, cerebral palsy and joint deformities [4,12,13,14,15].

Serial amnioinfusions with standard electrolyte solutions have seen little or no effects onto neonatal outcome probably because of the immediate loss of fluid after the intervention and possible negative effect of longtime saline solution on the fetus [4,14,16,17,18,19].

Tchirikov et al. performed the prolongation of the PPROM-to-delivery interval to 49 days in average patients with classic PPROM by using the intensive lavage of the amniotic cavity with Amnion Flush Solution (pump rate 100 mL/h) through a transabdominal catheter with an additionally designed anker system [4,12,20,21].

However, in recently published retrospective results from Japan, the authors did not improve the neonatal outcome by using a low rate of continuous amnioinfusion with Ringer’s lactate solution (40 mL/h) [14,15]. In the current German medical guidelines for the prevention of preterm birth, the amnioinfusion for the treatment of the PPROM is mentioned, but prospective randomized studies must be performed to clarify the positive effect of the continuous amnioinfusion on fetal survival [22].

## 2. Objectives

Principal research question: Does the intervention (continuous amnioinfusion—“flush out” method) reduce the adverse neonatal outcome of patients with PPROM with oligo/anhydramnion and prolong the pregnancy?

Primary hypothesis: The Amnion Flush Method will improve the neonatal outcome after PPROM between 20/0 and 26/0 weeks of gestation.

## 3. Materials and Methods

The full infrastructure of Coordination Center for Clinical Trials (KKS) of Martin-Luther-University Halle Wittenberg will be involved in ICH–GCP (Internation5al Conference on Harmonisation of Technical Requirements for Registration of Pharmaceuticals for Human Use–Good Clinical Practice) compliant data management and handling, randomization, safety management and monitoring of the study.

## 4. Study Setting

The study will be performed in 9 tertiary German perinatal centers, including five leading German universities (Figure 1).

ClinicalTrials.gov: NTC04696003. Filled circles: approved by local ethic committee.

Trial design:

Randomized multicenter trial; two-arm parallel.

Trial registration:

The trial was registered with ClinicalTrials.gov ID: NCT04696003 and German Clinical Trials Register: DRKS00024503.

Protocol version: Nr: KKSH163, 15 December 2020.

The TRIAL has obtained a permission of the Ethical Committee of the Halle University Medical Center (2020-185, 25 January 2021).

Funding:

The study is being funded by the German Federal Ministry of Education and Research (BMBF, No. 01KG2007).


**Eligibility criteria**


**Inclusion criteria:** single pregnancy (from 22/0 to 26/0 weeks of gestation), with classic PPROM with oligo/anhydramnion between 20/0 and 26/0 weeks of gestation. From the beginning of 22/0 weeks of gestation, the PPROM duration must be lower as of five days till randomization. The patients will receive RDS prophylaxis (respiratory distress syndrome) before randomization. The vertical deepest amniotic fluid depot must be <2 cm, and the amniotic fluid Index < 3. The PPROM will be confirmed by positive testing of the placental alpha microglobulin-1 and the amniotic fluid that is clearly lost (Figure 2, flow diagram).


**Exclusion criteria:**


The patients with fetal death, high rupture of membranes, pre-PPROM, premature labor, placental abnormalities, placental abruption, evidence of major structural or chromosomal abnormalities, indication for termination of pregnancy (e.g., HELLP syndrome, pathological CTG (if present), fetal bradycardia, eclampsia and umbilical cord prolapse), cervical insufficiency, placenta previa, signs of chorioamnionitis (maternal fever > 37.8 °C and two or more of the following criteria: uterine tenderness, malodorous vaginal discharge, maternal leukocytosis > 15,000 cells/mm^3^ (without corticosteroids), maternal tachycardia > 100 beats/min and fetal heart rate > 160 bpm) and CRP > 20 mg/dL (C reactive protein).


**Who will take informed consent?**


The authorized physicians of perinatal centers taking part in the trial will identify the potential trial participants with classic PPROM. Eligible women with classic mid-trimester PPROM will be asked for their written informed consent at the time of admission to the hospital, after having received oral and written information. Patients, who are not able to understand the informed consent will not be included. Additionally, for five years, all patients undergoing a fetal surgery therapy in hospital had filed a motion for a compassionate use from the ethical review committee. Recruitment for “flush out” method via a port-system is ethically sound for PPROM patients with oligo/anhydramnion, based on the available research evidence.

In case of complications, all equipment for emergency interventions will be easily accessible in the perinatal centers. An additional “safety net” for the finalization of the study and induction of delivery was established to avoid any complications.


**Intervention and control groups**



**Control group**


The PPROM patients of the control group receive the standard therapy related to the German Guidelines, including antibiotic therapy (Amoxicillin/Clarithromycin therapy or 7 days Amoxicillin and once Azithromycin 1 g per oz) or other antibiotics according to recommendations of the physicians from the microbiology department based on the culture results for at least 7 days [22]. There must be a daily blood analysis (hemoglobin, leucocytes, thrombocytes, CRP and interleucine-6) and a cervical smear every 5 days. The RDS prophylaxis must be performed before randomization.

**Amnion Flush interventional group** (Figure 3)

The PPROM patients of the Amnion Flush group will receive the standard therapy related to the German Guidelines, like the control group [22]. Additionally, after the amnioinfusion of 300 mL (Amnion Flush Solution Serumwerk AG, Bernburg, Germany) through the 22G needle (Figure 4A), under local anesthesia, an ultrasound-guided perinatal catheter with an anchor system (0.65 mm Diameter, CE 0481, PakuMed GmbH, Essen, Germany) will be inserted into the amniotic cavity through the 18G needle for the continuous amnioinfusion with 100 mL/h of Amnion Flush Solution, (Serumwerk Bernburg AG, Germany) until delivery (latest on beginning of 34/0 weeks of gestation) [4,12,13] (Figure 4B,C).


**Detailed description of the perinatal catheter implantation**


The catheter implantation will be performed in the operation room in aseptic conditions. Before the catheter implantation, the amnioinfusion with about 300 mL of Amnion Flush Solution will be performed by using a 22-gauge needle, under ultrasound control (Figure 4A).

The gynecologist must have experience with at least 100 amniocenteses because of the difficult conditions for the puncture related to anhydramnion.

The operation table allow for patients to be kept in the maximum Trendelenburg position in order to avoid the complete loss of infused Amnion Flush Solution because of PPROM. If patients complain about uterine contractions, the single short of Partusisten 25 µg will be slowly infused i.v.

The appropriate location for the catheter implantation (preferred area is the uterine fundus without placenta) will be identified by using ultrasound. Local anesthesia with Xylocaine 1% 10 mL will be injected [20]. The 18-gauge needle will be inserted into the amniotic cavity (Figure 4B).

The anchor catheter will be inserted through the needle. The 18 G needle will be removed. The catheter will be fixed up by using the sterile transparent foil (Figure 4C).

The Amnion Flush Solution will be infused by using standard medical pumps, at a rate of 100 mL/h (±20 mL). Daily ultrasound control must be performed to validate the intra-amniotic catheter position and the deepest deport of the amniotic fluid. The deepest pool of amniotic fluid should be stabilized (target 4 cm). The catheter should be replaced if indicated (e.g., delivery or catheter dislocation or plugging), but the latest should be every 30 days to avoid contamination and/or inflammation.


**Outcomes:**


**Primary outcome:** survival (binary yes/no) without major morbidities (severe bronchopulmonary dysplasia (BPD), and/or grade 3 or 4 intraventricular hemorrhage (IVH 3–4), and/or necrotizing enterocolitis (NEC) indicated for surgical intervention, and/or cystic periventricular leukomalacia (cPVL)) obtained one year after the delivery [23].

**Secondary outcomes:** Gestational age at delivery, duration of PPROM-to-delivery interval, perinatal mortality, incidence of histological affirmed chorioamnionitis, incidence of FIRS and/or maternal sepsis, cord-artery pH, APGAR scores, neonatal outcome at calculated age 36 weeks of gestation: duration of O_2_-supplementation, retinopathy of prematurity. Participants with surviving babies will send a prepaid validated outcome questionnaire at 12 months after the birth of their baby. The independent University Center of Clinical Studies of Martin-Luther-University Halle-Wittenberg will receive the copies of pediatric examinations performing monitoring and final statistical evaluation. Neurodevelopmental assessments, as well as the check-up, of the surviving children will be performed in tertian medical centers by trained pediatric specialists.


**Participant timeline**


The study site visits for monitoring the clinical trial will be carried out by chief and assistant monitors. During the clinical phase, the chief monitor will visit the medical centers every 6 months (see Table 1), thereby being available to be on staff, supervising the assistant monitors, and in order to address and solve any problems arising.

They will review the progress of recruitment and ensure participants’ rights and safety and the adherence to protocol regulatory compliance. During the query process by the KKS, which implies another month of study-site visits (beyond the end of recruitment), they will verify unclear data and obtain corrections. The chief monitor will also make sure that plenty of CRFs are available as staff, and questionnaires will be given to mothers to fill in prior to discharge from the hospital.


**Sample size and Statistic**


The sample-size calculation was based on the primary outcome (binary yes/no) survival without major morbidities (severe bronchopulmonary dysplasia (BPD), and/or grade 3 or 4 intraventricular hemorrhage (IVH 3–4), and/or necrotizing enterocolitis (NEC) and/or cystic periventricular leukomalacia (cPVL)).

The intervention is expected to prolong the gestation by around two weeks, resulting in an increase of the one-year survival rate without major morbidities after 27 gestational weeks of 67% (hazard = 0.0334). Based on Chen et al. (2016), the one-year survival rate without major morbidities after 25 weeks of gestation is around 36% (hazard = 0.0851) [23]. To be more conservative, the sample-size calculation was based on a hazard = 0.035 for the interventional group. Using a recruiting period of 21 months (four women per month) and an individual follow-up of one year, 34 women are needed in each group for a two-sided log-rank test with 5% significance level, a power of 80% and a loss-to-follow-up rate of 10% per year (hazard = 0.0088). Calculations were performed by PASS software.


**Recruitment:**


German gynecologists received flyers for the trial twice (2021 and 2022) in order to achieve adequate participant enrolment to reach the sample size. Additionally, a website for the study, via Google advert (keyword: “Amnion Flush Method”), was created.


**Assignment of interventions: allocation, concealment mechanism and implementation**


The randomization will be performed by using the randomizer of the center of the clinical studies of the Martin-Luther University Halle-Wittenberg, Germany (computer-generated random numbers). Before the randomization, the inclusion and exclusion criteria of the study will be printed by investigators into the randomizing table online. Matching criteria will be automatically analyzed, and the group of the study and the number will be generated online.


**Data collection and management**


The trial is registered with a full description at the German Clinical Trials Register (Deutsches Register Klinischer Studien, DRKS) and at Clinical Trials Register (ClinicalTrial.gov). The study’s protocol is published at the start of the trial.

The Coordination Center for Clinical Trials, University of Halle (Saale), is responsible for data management and archiving. The data will be managed and archived according to existing standards and regulations, in a data format, allowing long-term preservation and future reuse.

The data sheet will be completed when the baby is discharged from the hospital or after death. Any missing data will be reconciled by the chief investigator and trial administrator via contact with the PIs and examination of the hospital’s case notes. The data pack will be returned to the trial co-ordination centers after the baby is discharged or after death.

*P**ost-trial follow-up* is at 6 and 12 months postpartum, when questionnaires are sent to mothers’ home addresses for the assessment of subjective variables, as well as variables pertaining to neonatal medical care for the health/economic evaluation.

**CRF** will include data for recruitment and confirmation of eligibility, informed consent, baseline data, process, data during treatment, primary and secondary outcomes, other outcomes, safety variables, prognostic factors, various questionnaires, as well as data of the pediatric evaluations.

The KKS will receive the copies of pediatric examinations. Neurodevelopmental assessments as well as the check-up of the surviving children will be performed in medical centers by trained pediatric specialists.


**Monitoring**


The coordinating investigator is responsible for implementing and maintaining quality assurance and quality control systems with written SOPs to ensure that the trial is conducted and data are generated, documented and reported in compliance with the protocol, ICH–GCP and the applicable regulatory requirements. Tasks assigned to KKS Halle (see chapter 9, trial supporting facilities) will be fulfilled according to the written SOPs from KKS Halle and the German-wide KKS Network. Data from source documents will be transcribed onto the case report forms (CRFs) by the clinical research staff of each trial site. All source documents will be kept at the trial site, and no information in source documents about the identity of the patients will be disclosed. Pre-trial visits and on-site monitoring will be conducted by trained CRAs from KKS Halle. A total of 3–5 pre-trial visits will be performed to allow for the selection of 3 sites for participation. Based on the pre-trial visit reports, the PI and the steering committee will decide on the participation of the respective trial site. Because of previous on-site selection visits, for initiation of the study centers, conference calls (each site separately) will be considered. Initiation will be supported by .ppt presentations and study materials (investigator’s site file, protocol, informed consent and CRFs) earlier supplied to the sites. On-site monitoring will be performed on a risk-based approach and will be planned by means of regular visits from the beginning to the end of the trial to check the completeness of patient records, the consistency of entries on the CRFs and the adherence to the protocol and to Good Clinical Practice. Initially, the informed consent forms and the adherence to the inclusion and exclusion criteria will be checked. Thereafter, a formal check of captured information for completeness and plausibility will be performed, followed by a check of the correct transfer of data from the source data. Full (100%) source-data verification for the presence of informed consent and adherence to the inclusion/exclusion criteria will be performed. Data used for all primary and safety variables will also be 100% source-data verified. The specific extent of the monitoring and the source-data verification will be specified in the monitoring manual. Each trial site will be monitored once after the first trial participant has been enrolled and treated. The early monitoring shall ensure prompt intervention by the CRA or study center in case of any problems at the trial site, such as major inconsistencies in trial conduction and CRF completion. Another 7 (average, individual number depending on recruiting rate and quality of CRF completion) monitoring visits at each trial site will be conducted during the enrolment period. Every trial site will undergo a close-out visit by the CRA after the last participant at that site has finished the follow-up. The monitor’s access to the trial documents and medical records is ensured by the investigator’s agreement and the cooperation agreement between the sponsor and KKS Halle.

## 5. Statistical Analysis

The collected data will be analyzed by specialists of statistic of the Institute of Medical Epidemiology, Biostatistics and Informatics, Martin-Luther-University Halle-Wittenberg, Germany.


**Adverse event reporting harms**


The Data Safety Monitoring Board (DSMB) will follow the progress of the clinical trial; evaluate the safety and primary efficacy parameters; and will propose to the sponsor whether to continue, modify or stop a trial and provide the funding agency with information and advice. The DMSB will set up telephone conferences twice a year to evaluate current safety data. Therefore, the sponsor will provide up-to-date safety data, including a summary of AEs, line listing of SAEs and adverse device effects (ADEs). In case of a high number of severe unexpected events in between DMSB meetings or a case of a Suspected Unexpected Serious Adverse Reaction (SUSAR) or other medically important conditions, the DMSB will be informed by the sponsor immediately and may give advice for further procedures if required. An interim analysis is not planned.


**Stopping rules**


(a)For patients:

Placental abruption, indication for interruption of pregnancy (e.g., HELLP syndrome, pathological CTG, fetal bradycardia, eclampsia and umbilical cord prolapse), signs of chorioamnionitis (maternal fever > 37.8 °C and two or more of the following criteria: uterine tenderness, malodorous vaginal discharge, maternal leukocytosis > 15,000 cells/mm^3^ (without corticosteroids), maternal tachycardia > 100 beats/min and fetal heart rate > 160 bpm), CRP > 20 mg/dL, increased concentrations of interleukine-6 (>7 pg/mL), positive bacteriologic examination of the amniotic fluid, pos. Gram-stained sediment or IL-6 concentration > 2600 pg/mL or WBC > 30 mm^3^.

(b)For participating centers:

Inability to recruit the patients and complications during the catheter implantation, such as placental abruption, damage of the fetus or umbilical cord, fetal bleeding or death (audit after first complication, center ex in case 2 or more complications/year).


**For the whole trial:**


High range of complications in interventional group compared to the control group (we are going to audit the study every 3 months, discussing this point), unexpected adverse reactions to the medical products without correction possibilities, clear statistical benefit of “flush-out” treatment.


**
Frequency and plans for auditing trial conduct
**


The medical centers of clinical trial will undergo an audit twice a year by a study manager, KKS, Martin-Luther-University Halle-Wittenberg. To ensure the quality of data and study performance, the sponsor may conduct site visits by an independent auditor. An audit will be performed only after notification and arrangement with the investigator. An audit certificate will be issued as quality proof and has to be filed in the trial master file and as a copy in the investigator-site file.


**Ethics approval and consent to participate**


The study will be conducted in compliance with the Declaration of Helsinki and all amendments.

The TRIAL obtained permission from the Ethical Committee of the Halle University Medical Center (2020-185, 25 January 2021).

Written informed consent of the patients, as well as the parents, is required for enrolment in the study and can be withdrawn at any time, without reasons or disadvantages. Eligible women with classic mid-trimester PPROM will be asked for their written informed consent at the time of admission to the hospital, after having received oral and written information. Patients, who are not able to understand the informed consent will not be included. Recruitment for the “flush out” method via a port-system is ethically sound for PPROM patients with oligo/anhydramnion based on the available research evidence.

In case of complications, all equipment for emergency interventions will be easily accessible in the perinatal centers. Only hospitals which possess a Level 1 perinatal center in accordance with German law (tertian perinatal center) will receive permission to participate in this trial.

Any modifications to the protocol will be sent to the ethics committees before they are implemented within the study, and communication changes that impact the patients would require signing of a revised consent form.


**Trial Status:**


Protocol number KKSH163, Version No. 1.0, 27 October 2020.

Recruitment began: March 2021.

The approximate date when recruitment will be completed: January 2024.

## 6. Discussion and Review of the Literature

Mid-trimester preterm premature rupture of membranes (PPROM), together with oligohydramnios before 26 weeks, affects about 0.4–0.7% of all pregnancies. A high number of unreported cases should be expected in cases of PPROM prior to 22 weeks of gestation. Mid-trimester PPROM with anhydramnion is associated with a very high neonatal mortality rate, as well as an increased risk of long- and short-term severe neonatal morbidities; physical and developmental disabilities, including chronic respiratory disease, neurodevelopmental or behavioral effects (impairment of visual/hearing/executive functioning, global developmental delay and psychiatric/behavioral sequelae); and cardiovascular diseases [3,4,24]. The early birth due to PPROM with anhydramnion during the canalicular stage of the lung development between the 16th and 26th week of gestation leads to pulmonary hypoplasia [4,25]. Prolonged anhydramnion after PPROM is associated with a four-fold increased risk of composite adverse outcomes, including death, BPD, severe neurological disorders and severe retinopathy, when compared to an age-adjusted control group [8,26].

The causes of the mid-trimester PPROM are multifactorial, including local infiltration by bacteria with reaction of pro-inflammatory cytokines, pathologic anatomical remodeling of the amniotic membranes (contribution of MMPs), invasive procedures and fetoscopic surgery, genetic and iatrogenic factors, smoking, vaginal bleeding, etc. [4,7,27,28,29,30,31,32,33]. Inflammatory mediators likely play a causative role in both the disruption of fetal membrane integrity and activation of uterine contraction.

The diagnosis of PPROM is classically established by the identification watery leakage from the cervical canal, positive swab assay for placental alpha macroglobulin-1 and indigo carmine tests [4,34,35]. The management of the PPROM requires balancing the potential neonatal benefits from prolongation of the pregnancy and reduction of the adverse effects of newborn immaturity with the risk of intra-amniotic infection and its consequences for the mother and infant [2,4,22,36]. Close monitoring for signs of chorioamnionitis (e.g., body temperature, CTG, CRP, leucocytes, IL-6, procalcitonin and amniotic fluid examinations) is necessary to minimize the risk of neonatal and maternal complications.

Despite the fact that broad-spectrum antibiotics are routinely used in the therapy of PPROM based on the guidelines’ recommendations, it must be considered that the maternally applied drugs only hardly reach the place of bacterial colonization [4,19,22]. The amniotic membranes and the umbilical cord do not have an effective capillary net which would supply the surfaces with the antibiotics from the maternal circulation [37]. The placenta is a selective barrier for foreign substances, and the bioavailability among antibiotics is very low for most antibiotic agents [4,38,39,40,41,42,43]. On the other hand, the healthy fetus without any infection after PPROM probably does not need any longtime trans-placental antibiotic treatment that can lead to a possible change of fetal programming [4,12,13,44].

The Cochrane Library and the databases MEDLINE, PubMed, EMBASE, the Cochrane Library, DRKS and ICTRP were searched. To assess the methodological quality of the study, the articles are classified on the basis of the level of evidence (LoE).

Inclusion search criteria:

PPROM and amnioinfusion and treatment (non-diagnostic)

Exclusion criteria:

The exclusion criteria included surgery, diagnostic amnioinfusion, polyhydramnion, prophylactic amnioinfusion, term pregnancy, amnioinfusion during labor, relieving variable decelerations, transcervival amnioinfusion, dilute thick meconium-staining of the amniotic fluid, external cephalic version, amnioinfusion in connection with Amniocentesis, ex utero intrapartum therapy (EXIT) procedure and reviews older than 10 years; moreover, a publication was excluded if a more recent study from the same center or author was included. Only articles published in English or German were considered (Table 2).

A search of the literature in different search contexts revealed 82 hits, resulting in 28 relevant publications, consisting of reviews, randomized clinical trials (RCTs), non-randomized/observational clinical trials and case studies, outlining the state-of-the-art of serial or continuous transabdominal amnioinfusion in most cases by using standard infusion solutions that are used off-label in this regard. The positive effects of serial amnioinfusion were neonatal death RR 0.33 (0.14–0.66, moderate quality of evidence), neonatal sepsis/infection RR 0.26 (0.11–0.61, moderate QI), pulmonary hypoplasia RR 0.22 (0.06–0.88, low QI) and maternal puerperal sepsis RR 0.20 (0.05–0.84, QI moderate) [45].

Tranquilli et al. reported that serial transabdominal amnioinfusions (AIs) could prolong the latency period to a median of 21 days [46]. De Santis et al. found out that the patients with PPROM did not appear to demonstrate any benefit from this repetitive amniotic-fluid (AF) replacement (250 mL per intervention), as measured by post-procedure AFI, because fluid loss occurred within 6 h of instillation [19].

Locatelli et al. found that the serial AI could improve the neonatal outcome primarily by prolonging latency [16,47]. The patients with PPROM and oligohydramnios had a significantly shorter interval to delivery, lower neonatal survival of 20%, 62% of pulmonary hypoplasia and 60% rate of neurological handicap [16,47].

During a Canadian study [24], for infants delivered at 23 weeks of gestation, perinatal mortality was 89.9%, and of live-born neonates admitted to NICU neonatal, death occurred in 63.8%. Among survivors at discharge, the rate of severe brain injury was 44.0%, the rate of retinopathy of prematurity was 58.3% and the rate of any serious neonatal morbidity was 100% [24].

Roberts et al. concluded that serial transabdominal amnioinfusions showed no significant difference in the outcome of maternal and perinatal outcome. The perinatal mortality was 19/28 vs. 19/28. The positive fetal survival effect of serial AI was unfortunately counterbalanced by an increased risk of neonatal death: 14 neonates died in the AI group vs. 9 in the control group. It is possible that the use of saline solution for AI was inappropriate due to the large deviations from normal human amniotic fluid [17,18].

Thus, the accumulation of sodium and chloride would disturb the sodium–potassium pump located in the plasma membrane of human cells and could be influencing organs’ performance, e.g., cardiac, lung and brain. These adverse effects of the instillation fluid could explain the high mortality rate after AI in this study.

Moreover, repetitive puncture from AI method would increase the risk of separation of amniotic membrane from the uterus, abruption of the placenta and injury to the umbilical cord, thus causing a trauma to the fetus.

Van Kempen et al. performed a randomized controlled trial in women with PPROM at 16 0/7 to 24 0/7 weeks of gestation with oligohydramnios. Participants were allocated to transabdominal amnioinfusion weekly or no intervention until 28/0 weeks of gestation. The primary outcome was perinatal mortality. Perinatal mortality rates were 18 of 28 (64%) in the amnioinfusion group vs. 21 of 28 (75%) in the no-intervention group (relative risk 0.86, 95% CI 0.60–1.22, *p* = 0.39) [48].

**Search reviews:** The general benefit of serial AI results from the restoring of the amniotic fluid protecting the fetus from compression of the umbilical cord and mechanical compression of the thorax, allowing normal fluid flow into the lungs and thereby preventing fetal lung hypoplasia [19,46,49,50]. Additionally, it protects the fetus from postural deformities. The spreading of infection is inhibited by dilution or by the infused solution itself. Obviously, the latter effect should be enhanced by the application of continuous AI due to enhanced dilution by continuous flush-out of bacteria and inflammatory compounds. The procedural problems of performing continuous AI could be solved by the recent development of a subcutaneously implanted catheter system [12,13,20,51].

Our search of the literature revealed five reviews of relevance [4,45,49,50,52,53].

The review of Hofmeyr (2014) assessed the effects of AI with standard solutions versus no AI for third-trimester PPROM and considered five randomized controlled trials, three of them with transcervical AI and two with transabdominal AI, including data from 241 women in total [45]. For transabdominal AI, the evidence grade was moderate for neonatal death, neonatal sepsis/infection and maternal puerperal sepsis and low for pulmonary hypoplasia [45]. The authors concluded that the results are encouraging but more studies are needed.

The review of Kozinszky (2014) analyzed eight published studies (case series) about amnioinfusion for treatment in severe mid-trimester oligohydramnios after PPROM [53]. They concluded that amnioinfusion, compared to expectant management, significantly improves the perinatal outcome, reduces the risk of postural deformity and prolongs the pregnancy in severe second-trimester oligohydramnios in both idiopathic cases and those involving PPROM.

Preterm premature rupture of membranes (PPROM), together with oligohydramnios before 26 weeks, can delay lung development and can cause pulmonary hypoplasia; the critical interval is between week 16 and 26. Restoring the amniotic fluid volume by transabdominal amnioinfusion might prevent abnormal lung development and might have a protective effect for neurological complications, fetal deformities and neonatal sepsis. Therefore, van Teffelen (2013) aimed to assess oligohydramnios secondary to PPROM before 26 weeks but could not identify randomized controlled trials comparing transabdominal AI with no AI [49,50]. They concluded that further research focusing on PPROM occurring very early (<26 week) is needed [49].

In 2012, Porat et al. reviewed serial transabdominal amnioinfusion from RCTs and observational studies [52]. Two identified RCTs were on women with PPROM between 24 and 34 weeks; one quasi-randomized study included women with PPROM before 26 weeks [52]. The meta-analysis of the observational studies resulted in significant prolongation of latency, reduction of perinatal mortality, decreased rate of pulmonary hypoplasia and reduced risk of neonatal death. The same tendency was seen in the RCTs, but it was statistically non-significant. From two RCTs, a decreased rate of infectious complications was observed in the AI group. Porat et al. concluded that transabdominal amnioinfusion for early PPROM may improve early PPROM-associated morbidity and mortality rates, but more RCTs are needed to improve the evidence [52].

Three publications evaluated performance- and safety-relevant information of continuous amnioinfusion by using amniotic fluid solution (AFS). Furthermore, the first clinical experience with AFS was gained at the Martin-Luther-Universität Halle-Wittenberg by PI (MT) [13]. Eighteen case reports are provided describing the successful administration of AFS by continuous amnioinfusion. In total, the important clinical benefit of AFS is the prolongation of pregnancy, thereby improving the outcome and prognosis of the preterm infant. The complications are generally expected to be the same as those that can be caused by PPROM itself and by the procedure of amnioinfusion with standard solutions, but to a less extent. Complications due to the fluid composition are not reported, and, if any, they are expected to be mild or infrequent due to the physiologic composition of AFS. The positive effect of continuous amnioinfusion through the subcutaneously implanted perinatal port system with Amnion Flush Solution in “classic PPROM” less than 26 (28)/0 weeks’ gestation shows promise but has not been evaluated in a prospective randomized trial. Our feasibility study suggested that continuous amnioinfusion prolongs the mean duration of the PPROM delivery interval for 49 days (range, 9–69 days), improving neonatal outcome [4,12,13,20,21,51].

The patients treated with Jonosteril^®^ (Fresenius Kabi GmbH, Germany), Sterofundin^®^ and isotonic NaCl (B. Braun AG, Melsungen, Germany) or lactated Ringer’s solution (Baxter, Germany) reported significantly increased diuresis, probably triggered trans-planetary by a fetal response to increased NaCl, fluctuation of osmolality and missing microelements, also referred to as the “Salzgurken effect” [4]. The electrolyte solutions for the continuous amnioinfusion were replaced in 2012 by artificial amniotic fluid (now Amnion Flush Solution, CE0483, Serumwerk AG, Bernburg, Germany) without any adverse events [4,12,13].

However, two groups from Japan in recently published retrospective studies (2020) were not able improve the neonatal outcome by using the low-rate amnioinfusion with Ringer’s lactate solution (40 mL/h) [14,15]. The authors mentioned the high frequency of catheter dislocation (60%) and lack of pump infusion in many cases [14,15]. The missed positive effect of the prolongation of the PPROM-delivery interval by two weeks onto healthy neonatal survival in these studies could be explained by “Salzgurken effect” of *off-label used* saline solution for a long time with a very high deviation of many parameters compared to human amniotic fluid (Table 3) [4].

Fetal skin in second trimester is still very thin and permeable. It is a matter of common knowledge that the fetus swallows and inspirates/expirates amniotic fluid. Gilbert and Brace indicated that the fetus swallows 200–250 mL/kg/day amniotic fluid [54]. Continuous amnioinfusion with normal saline solution significantly increased plasma Na^+^ and Cl^-^ concentrations in fetal sheep [55]. The amniotic fluid is a very complex hypoosmotic solution with an alkaline pH; low concentration of the elements Cl^−^, K^+^ and Na^+^; and the presence of trace elements, surfactants and many other substances [56].

Trace elements such as copper, zinc and selenium are needed for many enzymes and cofactors. They function as free radical scavengers and help to protect from oxidative stress. Several antioxidant enzymes are metal-dependent, such as Cu-Zn superoxide dismutase (SOD) and selenium-dependent glutathione peroxidase (GSH-Px) [57]. It could be shown that a deficiency in these trace elements may result in preterm birth and in clinical signs of the fetus such as poor growth, immature gastrointestinal tract and disturbed development of organs [58]. The zinc content of the amniotic fluid has an important metabolic role, and a positive correlation between zinc concentration of the amniotic fluid and fetal weight and length was found. The zinc content of the amniotic fluid increases significantly during the third trimester [59].

Selenium deficiency in females is associated with infertility and spontaneous abortion, suggesting a role for selenium-requiring proteins during embryonic development [60]. The amniotic selenium level significantly decreases with the progress of normal pregnancy. This might be in part attributed to the greater demand for the nutrient by the fetus during late pregnancy [61].

Therefore, the AFS should contain the trace elements copper, zinc and selenium in concentrations, as reported for natural human amniotic fluid. Obviously, the concentrations of copper and selenium in AFS match well with the reported concentrations from the literature. The reported zinc concentrations in amniotic fluid from different studies showed a large variation (0.3 to 7.6 µmol/L*). The concentration of 2.6 µmol/L in AFS reflects this variation and is justified due to the fact that zinc is essential and highly required by the fetus [62].

The change of physiological fetal surroundings for a long period by using amnioinfusion with simple electrolyte solutions could destroy, in all probabilities, the fetal programming. Hypernatremia-induced osmotic demyelization disorders and central pontine and extrapontine myelinolysis are well-known [63]. Chhabra et al. described the extra-pontine myelinolysis induced by hypernatremia [64]. The combination of the very immature brain–blood barrier of the fetus with the permeable skin and swallowing of a relatively large amount of electrolyte solution deteriorates the fluctuations of the NaCl concentration of the fetal brain. This could explain the high neonatal mortality rate after serial amnioinfusion in the Roberts et al. study and very moderate success of continuous amnioinfusion from using Ringer’s lactate solution (inappropriate to human amniotic fluid) described in the retrospective Japanese studies [14,15,17,18].

In summary, the standard treatment options of the classic mid-trimester PPROM with oligo/anhydramnion, including RDS prophylaxis and broad-spectrum antibiotics, hardly improve the neonatal outcome. The extension of the standard therapy by continuous amnioinfusion—the Amnion Flush Method—could improve the neonatal survival without major morbidities. In 2020, the German Federal Ministry of Education and Research funded this prospective multicenter randomized study investigating the effect of continuous amnioinfusion in patients with classic PPROM with oligo/anhydramnion between 22/0 (20/0) and 26/0 WG with Amnion Flush Solution (100 mL/h, 2400 mL/d) through a transcutaneous perinatal catheter with an anchor system compared with the standard PPROM therapy (ClinicalTrials.gov: NTC04696003).

## Figures and Tables

**Figure 1 life-12-01351-f001:**
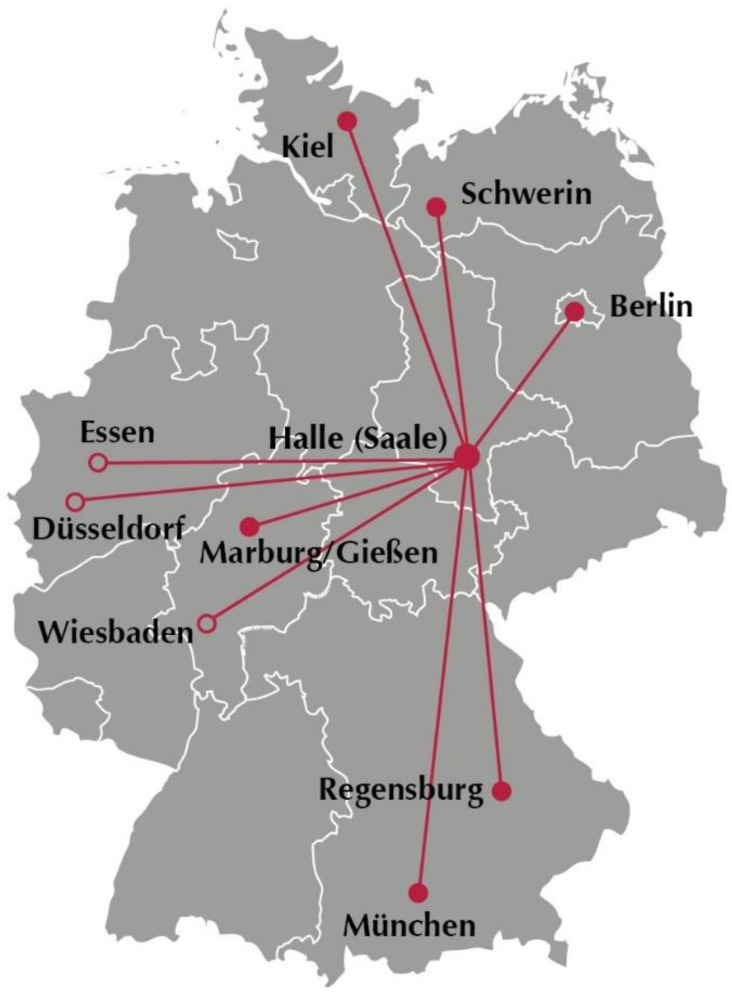
German tertian perinatal centers taking part in the TRIAL.

**Figure 2 life-12-01351-f002:**
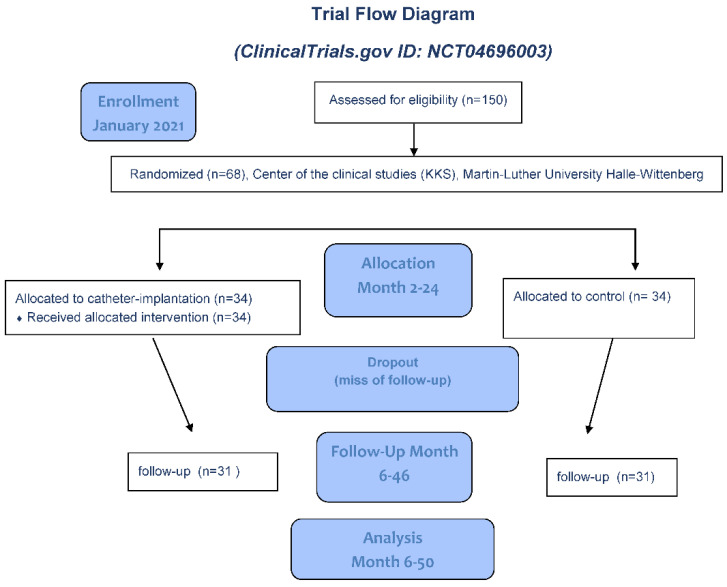
Flow diagram of the TRIAL.

**Figure 3 life-12-01351-f003:**
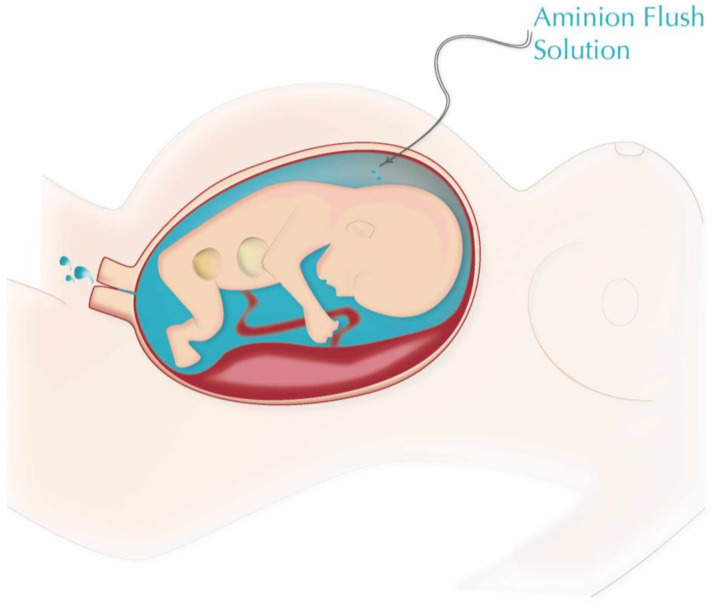
The schema of Amnion Flush Method.

**Figure 4 life-12-01351-f004:**
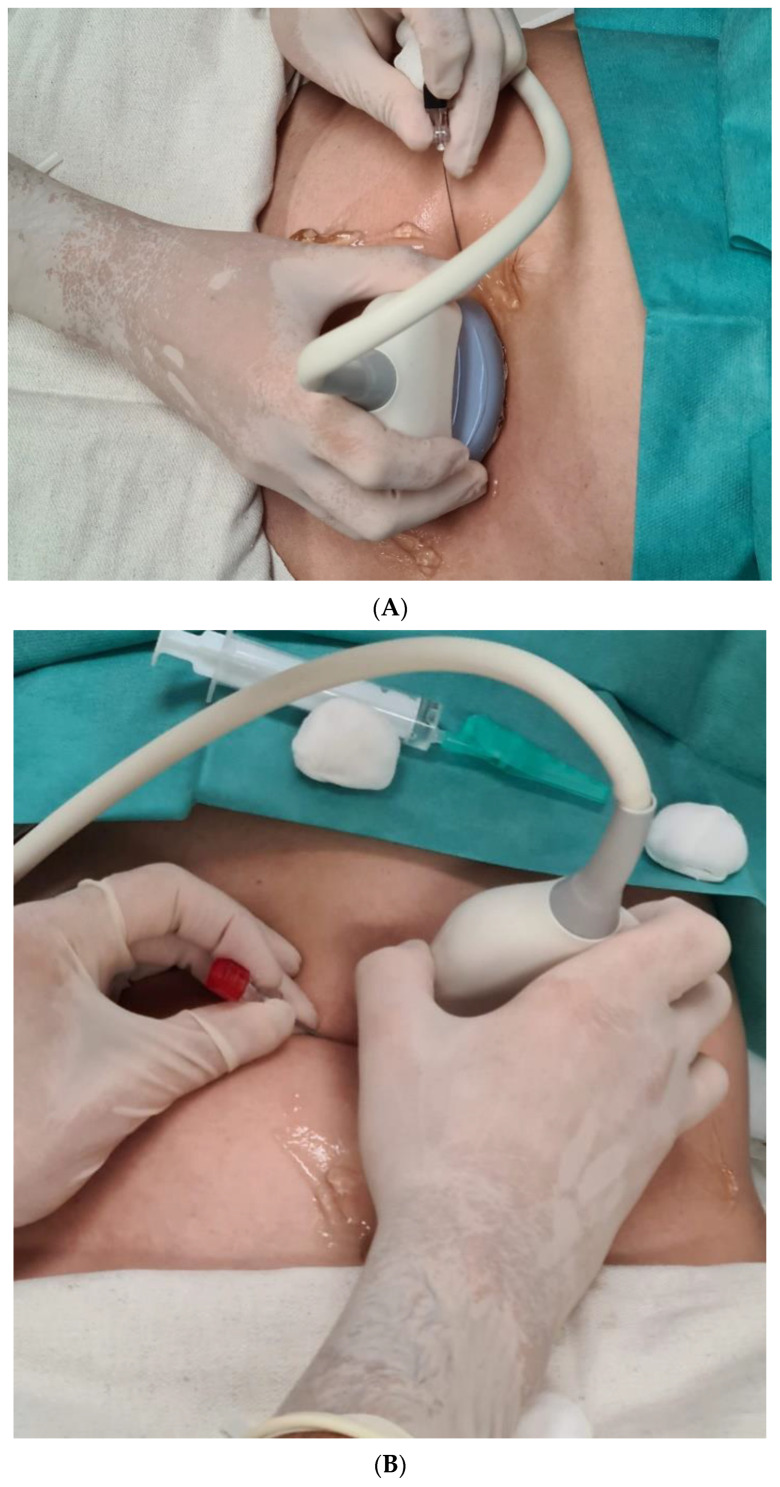
(**A**) Amniocentesis with 22G needle and amnioinfusion 300 mL of Amnion Flush Solution (Serumwerk AG, Bernburg, Germany). (**B**) Amniocentesis with 18G needle under local anesthesia and introduction of the catheter. (**C**) The intra-amnial catheter (arrow) is covered by Tagaderm film. The rate of intra-amnial infusion is 100 mL/h (Amnion Flush Solution, Serumwerk AG, Bernburg, Germany).

**Table 1 life-12-01351-t001:** Time points of visits and procedures every 6 months (x) and per month (xx).

	Screening Day −5–0	Randomization Day 0	Therapy Day 0–85 (max)				Follow-Up Day 01–365 (± 30 d) Postpartum			
			Daily	Every 7 d (±1 d)	As Needed	Dies Natus	1 d Postpartum	Day of Release from Hospital	Calc. GA 36 (0) (±3 d)	365 d Postpartum (±30 d)
**all subjects**										
**informed consent**	**x**									
**physical examination**	**x**							**x**		
**ultrasound**	**x**	**x**	**x**							
**blood**	**x**	**x**	**x**			**x**	**x**			
**biochemistry**	**x**	**x**	**x**			**x**	**x**			
**bioch. PPROM-test**	**x**									
**RDS prophylaxis**	**x**									
**randomization**		**x**								
**syst. antibiosis**	**x**		**x**		**x**					
**histology of chorioamnionitis**						**x**				
**histology of placenta**						**x**				
**flush-out group subjects**										
**catheter implantation/change**		**x**			**x**					
**amnion flush**		**x**	**x**							
**microbiology (amniotic fluid)**				**x**	**x**					
**neonates**										
**smear test**						**x**				
**APGAR-Score**						**x**				
**neurological examination**						**x**	**x**	**x**	**x**	**x**
**physical examination**										
**vital signs**						**x**	**x**	**x**	**x**	**x**
**weight/height**						**x**	**x**	**x**	**x**	**x**
**AEs**						**x**	**x**	**x**	**x**	
**Bayley III Scale**										**x**

GA, gestational age; RDS, respiratory distress syndrome; AE, adverse event.

**Table 2 life-12-01351-t002:** Search and results.

Search No.	Search Term 7 July 2022	Results	Relevant
1	((preterm premature rupture of membrane) or PPROM) and amnioinfusion	55	24
2	amnioinfusion and prolong and (pregnancy or (time to delivery))	5	2
3	(PPROM) and (continuous amnioinfusion)	9	6
4	continuous amnioinfusion	23	6
Total		82	28

**Table 3 life-12-01351-t003:** Comparison between the Amniotic Flush Solution and standard saline solutions in relation to the human amniotic fluid.

	Human Amniotic Fluid *	Amnion Flush Solution	Isotonic NaCl	Sterofundin
mmol/L Meal Values		Solution for Continuous Amnioinfusion	Solution for Infusion	Solution for Infusion
Sodium	134.8	132.3	154	140
Potassium	3.9	3.9	-	4.0
Magnesium	0.57	0.57	-	2.5
Calcium	1.9	1.6	-	-
Chloride	110	110.7	154	106
Lactate	9.1	9.1	-	45
Gluconate	0–18 µmol/L	3.20	-	-
Bicarbonate	16.9	16.9	-	-
Phosphate	1.07	0.35	-	-
Citrate	0.35	0.35	-	-
Trace elements				
	Concentration, µmol/L (mean values)			
Copper	2.51	2.52	-	-
Selenium	0.17	0.15	-	-
Zinc	0.3–7.62	2.6	-	-
pH and Osmolarity				
Theor. Osmolarity (mOsm/L)	273	278.9	309	299
pH	7.0–8.0	7.0–8.0	5.0–7.0	4.5–7.5

*: mean values US 9,072,755 B2, EP 2661 308 B1.

## Data Availability

Not applicable.

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
