# Peer review of "Treatment of Classic Mid-Trimester Preterm Premature Rupture of Membranes (PPROM) with Oligo/Anhydramnion between 22 and 26 Weeks of Gestation by Means of Continuous Amnioinfusion: Protocol of a Randomized Multicentric Prospective Controlled TRIAL and Review of the Literature"

_life, 2022, doi:10.3390/life12091351_

Round 1

Reviewer 1 Report

Dear Authors

This is a remarkable article with very important results in the field of obstetrics, as well as neonatology.

I just have to point out that you should add footnotes under the tables to explain what the numbers and symbols mean.

Also, you have to pay attention to the font especially in the tables because it shows inconsistency in some places

Regards

Author Response

Reviewer: 1

Dear Authors

This is a remarkable article with very important results in the field of obstetrics, as well as neonatology.

I just have to point out that you should add footnotes under the tables to explain what the numbers and symbols mean.

  • Is done

Also, you have to pay attention to the font especially in the tables because it shows inconsistency in some places

  • Thank you very much for your suggestions. The tables have been corrected.

Reviewer 2 Report

Dear Authors,

interesting conception

my comments:

1. "termination of pregnancy" in my opinion too negative soundness

2.  exclusion critetria -You wrote about pathological CTG- between 22-26 weeks of gestation- really? According the knowledge, which I have CTG over 26 weeks of gestation.

3. In my opinion all abbreviatinos should be described e.g. "Th"- about what did you think?

4. too much self-citations.

5. text should be written in the same style- font, size.. etc.

Author Response

Reviewer: 2

Dear Authors,

interesting conception

my comments:

  1. "termination of pregnancy" in my opinion too negative soundness

– “Termination of the pregnancy” has been exchanged by “interruption of the pregnancy”.

  1. exclusion critetria -You wrote about pathological CTG- between 22-26 weeks of gestation- really? According the knowledge, which I have CTG over 26 weeks of gestation.

– You are right; the sufficient evaluation of the CTG is possible after 26+0 week of gestation. However, in Germany some gynecologists perform the CTG control since 24+0 WG.

– We have corrected the sentence “(in present)”.

  1. In my opinion all abbreviatinos should be described e.g. "Th"- about what did you think?

Thank you for this advice. Th: thrombocytes. We have reduced the abbreviations of the manuscript.

  1. too much self-citations.

You are right, sorry. It is very difficult to avoid the self-citation in this Amnion Flush paper because I really developed this method of Amnion Flush for the PPROM patients 2008 at the University of Mainz, Germany.

I received an “Innovation Award Halle 2017” for these innovations.

I am a winner of the Hugo Junkers Award 2017, Germany (1st Place) in the category: The most Innovative Project and Applied Research. Innovations’ title: “Artificial amniotic fluid and perinatal port-system for the treatment of preterm premature rupture of the membranes”.

I sincerely hope for your understanding.   

  1. text should be written in the same style- font, size.. etc.

It is done

Reviewer 3 Report

This  study explores a new interesting field.

The protocol is well organized with appropriate language and a valid scientific structure.

Author Response

Reviewer: 3

Comments to the Author

This  study explores a new interesting field.

The protocol is well organized with appropriate language and a valid scientific structure.

Many thanks!

Round 2

Reviewer 2 Report

Dear Authors,

I accept your response.